# Toxic Effects on Thyroid Gland of Male Adult Lizards (*Podarcis Siculus*) in Contact with PolyChlorinated Biphenyls (PCBs)-Contaminated Soil

**DOI:** 10.3390/ijms23094790

**Published:** 2022-04-26

**Authors:** Rosaria Sciarrillo, Alessandra Falzarano, Vito Gallicchio, Aldo Mileo, Maria De Falco

**Affiliations:** 1Department of Science and Technologies, University of Sannio, 82100 Benevento, Italy; alfalzarano@unisannio.it; 2Vascular Surgery, Hospital of National Importance San Giuseppe Moscati, Via Contrada Amoretta, 83100 Avellino, Italy; vitogallicchio@gmail.com; 3Department of Biology, University of Naples “Federico II”, 80126 Naples, Italy; aldo.mileo@unina.it (A.M.); madefalco@unina.it (M.D.F.); 4National Institute of Biostructures and Biosystems (INBB), 00136 Rome, Italy; 5Center for Studies on Bioinspired Agro-Environmental Technology (BAT Center), 80055 Portici, Italy

**Keywords:** thyroid hormones, reptiles, endocrine-disruptor chemicals, lizard, hepatic deiodinase II

## Abstract

Skin exposure is considered a potentially significant but little-studied pathway for PolyChlorinated Biphenyls uptake in terrestrial reptiles. In this study, a native Italian lizard, *Podarcis siculus*, was exposed to PCBs-contaminated soil for 120 days. Tissues distribution of PCBs, thyroid hormone levels, and thyroid histo-physiopathology were examined. The accumulation of PCBs in skin, plasma, liver, kidney, and brain were highest at 120 days. The alteration of triiodothyronine (T_3_) and thyroxine (T_4_) levels after different concentrations and times to exposure of PCBs was accompanied by the changes in the hormones involved in the hypothalamus-pituitary-thyroid (HPT) axis, namely Thyrotropin Releasing Hormone (TRH) and Thyroid Stimulating Hormone (TSH). Moreover, hepatic levels of deiodinase II (5′ORDII) and content of T_3_ were positively correlated to exposure to PCBs. These results indicated that in lizards, PCBs exposure through the skin has the potential to disrupt the thyroid endocrine system. Overall, the observed results indicate that PCBs could be associated with changes in thyroid homeostasis in these reptiles, through direct interactions with the metabolism of T_4_ and T_3_ through the HPT axis or indirect interactions with peripheral deiodination.

## 1. Introduction

Polychlorinated biphenyls (PCBs) are ubiquitous and persistent organochlorine compounds widely spread in the global environment that pose a danger to human health and ecological systems. These pollutants, due to their high toxicity and lipophilic property, can accumulate in soil, plants and animals. Moreover, because of the absence of adequate metabolic pathways in the organisms, these substances tend to bioaccumulate along the trophic chains [1,2]. They are classified as “Endocrine-Disruptor Chemicals” (EDCs) because they can interact with several functions of the endocrine system and are particularly able to affect the thyroid gland [3,4]. PCB congeners are similar in TH structure, resulting in direct interaction with Thyroid Hormones (TH) receptors (TR) leading to the destruction of the normal thyroid homeostasis [4,5,6,7] but the interference between PCBs and TH levels is controversial. 

PCBs are frequently found in aquatic matrices, and it has been reported that they alter thyroid histopathology and interfere with metabolism, activity, and level of TH in fish [8,9,10].

Previous in vivo studies have shown that the effects of Aroclor 1254 (2, 2′, 3, 3′, 4-pentachlorobiphenyl) on growth retardation and metamorphosis inhibition in fish *Paralichthys olivaceus,* were correlated with a decrease in T_4_ and T_3_ levels [11]. Similar results were found by Couderc et al. [12] in wild European eels (*Anguilla anguilla*) where PCBs and polybrominated diphenyl ethers (PBDEs) cause changes in thyroid homeostasis, both with direct and/or indirect interactions with peripheral deiodination, T_4_ metabolism and mechanisms involved in thyroid-stimulating hormone (TSHβ), deiodinase 2 (Deio 2), and hepatic vitellogenin (Vtg) gene transcription.

In vivo studies using rodents and monkeys have shown that exposure to PCBs induces a decrease in circulating TH levels [13,14], while different effects on serum TH levels in dogs and cats following a single administration of a mixture of 12 PCB congeners were detected probably due to differences in the metabolic capacity of PCBs and TH between animals [6]. Total T_4_ and T_3_ levels decreased and free T_4_ levels increased in PCB-exposed dogs while no change in TH levels was observed in PCB-exposed cats. Different effects on serum TH levels in dogs and cats following a single administration of a mixture of 12 PCB congeners are likely to be attributed to differences in metabolic capacity for PCBs and THs between the animals [6].

Reptiles are the most diverse groups more sensitive than terrestrial vertebrates to contaminants as they can be exposed to environmental pollution through various routes, including the ingestion of contaminated food, polluted water or soil [15,16,17,18,19]. Although ingestion of contaminated food is probably one of the most important avenues, contaminants could also enter reptiles through skin exposure [20,21,22,23] to polluted soil. Previous research has suggested that dermal exposure may be more important than previously considered for reptiles [24,25,26,27]. In fact, Weir et al. [27] recently demonstrated that reptile skin permeability toward pesticides is, in fact, low. A previous study reported that lizards exposed to the same quantities of pesticides via oral and dermal routes resulted in similar residue values [25]. Thus, dermal uptake should not be disregarded [28,29].

In recent decades, the increase in environmental pollution together with climate change and habitat loss, are among the causes responsible for the decline of reptiles [30,31,32,33,34]. Lizards can be used as a representative reptilian group to determine the toxicity of PCBs, thus providing the information needed for ecotoxicology risk assessment of organic pollutants. *Podarcis siculus* was selected as an experimental model in this study because it is the most abundant species of lizards in Campania [19,35], it is easily bred in the captivity, and its favorite habitat (fields and areas up to 1800 m) brings it into close contact with the soil.

Furthermore, the morphophysiology of the thyroid gland of *Podarcis siculus* has been well documented [19,36] and given the lack of data on the relationship between soil contaminated with PCBs and accumulation in lizards, our study was designed to evaluate the accumulation of PCBs in tissue matrices (plasma, liver, skin, brain and kidney) of lizards that live on PCBs-contaminated soil. Moreover, we also investigated the interference of PCBs on thyroid hormones (THs) and Thyrotropin Releasing Hormone (TRH), Thyroid Stimulating Hormone (TSH) and morphology of the thyroid gland of lizards that live on PCBs-contaminated soil. We also investigated hepatic T_3_ and T_4_ contents and deiodinase types II (5′ORDII) activity to identify the effect of PCBs on the liver, which is the principal target organ of THs. 

## 2. Results

### 2.1. Signs of Toxicity, Animal Mortality and Body Weight

Signs of toxicity, mortality and body weight of the animals were continuously monitored throughout the experiment after 0, 30, 60, 90 and 120 days. Early signs may include hind limb discoordination and a change in breathing frequency and effort. Hind limb paralysis begins as mild to pronounced incoordination and weakness. As the paralysis rises, the animal becomes unable to move. The respiratory anomaly is of greater prognostic importance than the limb paralysis. The respiratory rate may initially increase but, as the contamination progresses, becomes slower and obviously tiring, especially at the end of the experiment (120 days).

Mortality of 6% and 12% was recorded in low-dose PCBs (Group A) and medium-dose PCBs (Group B), respectively; moreover, the highest percentage of mortality (33%) was observed in lizards treated with high-dose PCBs (Group C) and with PCB mix (Group D) (Table 1).

Interestingly, the weight of the specimens decreased as the days of treatment increased and this reduction was more evident in the specimens treated with higher doses of PCBs at 120 days (Group C, *p* < 0.001) (Table 2). There was no decrease in the appetite of lizards that were fed normally during exposure to PCBs. Therefore, lizard growth was observed to be significantly affected by exposure to PCB contaminated soil at 90 (9 g ± 0.05) (*p* < 0.001) and 120 (8 g ± 0.05) days, mainly in Group C compared to control specimens not exposed to PCB (17 g ± 0.05) (*p* < 0.001). The same result is evident with intraperitoneally exposure to PCBs (Group D), where the body weight of the lizards is significantly reduced at 90 (9 g ± 0.05) (*p* < 0.001) and 120 days (8 g ± 0.05) (*p* < 0.001).

### 2.2. Plasma Levels of Hormones Belonging to the HPT Axis

There were no significant differences between the plasma levels of hormones belonging to the HPT axis of the lizards housed for 120 days in nonpolluted terraria (time zero controls) and lizards treated intraperitoneally with a single dose of corn oil (Control group).

Data obtained showed that thyroid hormonal changes are related to the PCB exposure dose. Lizards treated with low-dose PCBs (Group A) showed an increase in TRH levels that ranged from 2.35 ± 0.05 μUI/mL in the control group to 4.29 ± 0.08 μUI/ mL (*p* < 0.001) (Figure 1a). In contrast, TSH plasma level decreased from 4.58 ± 0.05 μUI/mL (Control group) to 3.22 ± 0.02 μUI/mL (Group A) (Figure 1b) (*p* < 0.001). In accordance with the reduction of TSH levels, T_3_ and T_4_ values were also reduced, respectively from 4.22 ± 0.05 ng/mL (Control group) to 3.58 ± 0.02 ng/mL (Group A) (Figure 1c) (*p* < 0.001) and from 7.54 ± 0.10 ng/mL (Control group) to 6.16 ± 0.05 ng/mL (Group A) (Figure 1d) (*p* < 0.001).

A further increase in TRH levels and decrease in TSH levels was seen in both medium-dose PCBs (Group B) and high-dose PCBs (Group C) (Figure 1a). Specifically, TRH levels reached the values of 5.12 ± 0.05 μUI/mL (Group B) (*p* < 0.001) and 6.32 ± 0.05 μUI/mL (Group C) (*p* < 0.001). Contrariwise, TSH levels were reduced to 2.75 ± 0.01 μUI/mL (Group B) (*p* < 0.001) and 1.96 ± 0.02 μUI/mL (Group C) (*p* < 0.001) (Figure 1b). Similarly, T_3_ reached to the value of 2.13 ± 0.05 ng/mL (Group B) (*p* < 0.001) and 1.01 ± 0.02 ng/mL (Group C) (*p* < 0.001) (Figure 1c) and T_4_ reached to the value of 3.25 ± 0.02 ng/mL (Group B) (*p* < 0.001) and 1.25 ± 0.05 (Group C) (*p* < 0.001) (Figure 1d).

These patterns of levels of hormones plasma were similar in the male lizards treated intraperitoneally with a single dose of mix PCB and sacrificed 24 h after injection (Group D). In fact, in Group D, TRH level showed almost three-fold increase when compared to control group (6.85 ± 0.10 μUI/mL vs. 2.35 ± 0.05 μUI/mL) (*p* < 0.001) (Figure 1a). In contrast, TSH plasma level decreased from 4.58 ± 0.05 μUI/mL (control group) to 1.85 ± 0.02 μUI/mL (Group D) (*p* < 0.001) (Figure 1b). In accordance with the reduction of TSH levels, T_3_ and T_4_ values were also drastically reduced, reached respectively from 4.22 ± 0,05 ng/mL (control group) to 1.11 ± 0.01 ng/mL (Group D)) (*p* < 0.001) (Figure 1c) and 7.54 ± 0.10 ng/mL (control group) to 1.52 ± 0.01 ng/mL (Group D)) (*p* < 0.001) (Figure 1d).

### 2.3. Hepatic Thyroid Hormones Content and 5′ORD (Type II) Monodeiodinase Activity

Lizards housed in contaminated soil-filled terraria with PCBs had a higher concentration of T_3_ and a lower concentration of T_4_ in the liver with an increased 5′-ORD (type II) activity (Figure 2).

In detail, an increase in 5′ORD (type II) activity was observed in lizards treated with both medium-dose PCBs (Group B)(5.96 ± 0.10 pM T_3_/g/h) (*p* < 0.001), and high-dose PCBs (Group C) (6.13 ± 0.12 pM T_3_/g/h) (*p* < 0.001) and the highest increase was recorded in lizards treated with PCB mix (Group D, 7.32 ± 0.09 pM T_3_/g/h) (*p* < 0.001) (Figure 2a).

Similarly, hepatic T_3_ content increased reached from 3.52 ± 0.04 ng/mg of tissue fresh weight (control Group) to 4.72 ± 0.05 ng/mg of tissue fresh weight (Group B) (*p* < 0.001), 5.65 ± 0.02 ng/mg of tissue fresh weight (Group C) (*p* < 0.001) and 5.65 ± 0.05 ng/mg of tissue fresh weight (Group D) (*p* < 0.001) (Figure 2b). On the contrary, T_4_ hepatic contents decreased in all treated groups reaching the minimum value of 1.12 ± 0.05 ng/mg of tissue fresh weight in Group D (*p* < 0.001) (Figure 2c).

### 2.4. Determination of PCBs in Plasma and Tissues (Liver, Skin, Brain, and Kidney)

After 120 days of PCB exposure, the concentrations of PCBs in different tissues of lizard were determined in all groups and the results are listed in Figure 3. The highest concentration was observed in Group C and the order of concentration was skin (Figure 3a), plasma (Figure 3b), followed by the liver (Figure 3c), kidney (Figure 3d), and brain (Figure 3e). The concentrations of PCBs in the skin, plasma, liver, kidney and brain were 299.29 ± 25 ng/ml^−1^, 2.69 ± 52 ng/ml^−1^ and 216.04 ± 30 ng/ml^−1^, 53.26 ± 2.5 ng/ml^−1^ and 29.99 ± 5 ng/ml^−1^ respectively in the Group C (*p* < 0.001). These findings might be related to the fact that PCBs were absorbed from the soil by the skin and entered blood circulation. Therefore, we assumed that in lizards the PCBs would be absorbed by the skin and then distributed to other organs by blood circulation. PCBs concentrations in the skin were always higher than those in other organs in the different exposure groups. The orders of PCBs concentrations were as follows: skin > plasma> liver> kidney > brain both in the low-dose PCBs (Group A) (*p* < 0.001) and in the high- dose PCBs (Group C) (*p* < 0.001).

For Group D lizards, the lowest concentration of PCBs was found in the skin (16.02 ± 1.05 ng/mL^−1^) (*p* < 0.001) and brain (22.5 ± 1.15 ng/mL^−1^) (*p* < 0.001) respectively. The concentrations of PCBs were significantly higher in the liver (276.02 ± 8.055 ng/mL^−1^) than in the plasma (220.95 ± 5.50 ng/mL^−1^) (*p* < 0.001). Therefore, in this study, the accumulation of PCBs in the different tissues of lizards can be correlated with the route of exposure; PCBs slowly enter the systemic circulation mainly through skin permeation. Moreover, further intake of PCBs could occur through either accidental or intentional ingestion of the soil in lizards.

### 2.5. Thyroid Glands Histology

The thyroid histopathological observation of the lizard at the end of continuous exposure was shown in Figure 4. *Podarcis siculus* thyroid gland crosses the center of the trachea transversely, has a ribbon-like appearance and consists of follicles that are connected to each other by an interfollicular connective tissue that is traversed by blood vessels. The gland is wrapped in a capsule of superficial connective tissue that branches out and forms a network that surrounds the follicles. Each follicle is enveloped in a cuboidal epithelium, formed by thyrocytes and containing a medium-sized colloidal mass [18]. In the control group, the thyroid showed a normal histological aspect. Colloidal-filled follicles surrounded were observed or cubic epithelial cells. The follicles were mainly elliptical (Figure 4a). The height of follicular epithelial cells in the control and exposure groups are shown in Table 3.

Compared to the control group (22.1 ± 0.02 µm) the height of the follicular epithelial cells in groups A, B and C varied 10.6 ± 0.03 µm, 6.61 ± 0.01 µm and 3.32 ± 0.05 µm (*p* < 0.001) respectively. Morphological analysis of the thyroid gland in Group A (Figure 4b), Group B (Figure 4c) and mainly Group C (Figure 4d) showed, in addition to reduction in the height of the follicular epithelium, also a total retroaction of the colloid with few reabsorption vacuoles. Moreover, the nuclei of the thyrocytes were small and elongated with dense chromatin and a greatly reduced cytoplasm. The thyroid gland showed very evident signs of a poor functional activity. In the lizards treated intraperitoneally with a single dose of PCB mix (Group D), the thyroid showed a reduced height of the follicular epithelium (3.15 ± 0.04 µm) (*p* < 0.001) and a retracted colloid with few reabsorption vacuoles (Figure 4e).

## 3. Discussion

The present study identified significant changes in the Italian lizard thyroid gland caused by exposure to PCB-contaminated soil. Lizards are exposed to environmental contamination in different ways; while, the ingestion of contaminated food, water or soil is the most important way, the skin exposure to polluted soil is another important route for the uptake of PCBs in *p. siculus*. In fact, in the exposure group’s lizards (Group C), the concentrations of PCBs were highest in the skin. This might be because the skin was in close contact with the contaminated soil. Therefore, our results show that PCBs were absorbed from the soil through the skin and entered the bloodstream. Interestingly, the concentrations of PCBs are lowest in the brain and kidneys, suggesting that the kidney has strong detoxification abilities that allow it to metabolize quickly and eliminate the xenobiotic; while, in the brain, perhaps, the xenobiotic is unable to penetrate completely thanks to the blood-brain barrier.

The histopathology of the thyroid gland, plasma thyroid hormone levels, the hypothalamus-pituitary-thyroid (HPT) axis and the peripheral tissues have been measured and correlated in order to evaluate the thyroid disrupting effects of PCBs.

In this study, in the PCBs exposure Group C, follicular epithelial cells showed a low columnar shape and their height decreased compared to the control group. This indicated that the follicular epithelial cells were inactive. This result was consistent with a significant decrease in T_3_ concentration in lizard’s plasma in all groups exposed to PCBs. In addition, it was noted that in the exposed group PCBs, the proportions of follicles decreased significantly compared to the control group. These data showed that exposure to PCBs not only caused hypothyroidism, but also caused morphological damage to the lizard’s thyroid gland. However, we observed reabsorbed vacuoles in the follicles (Figure 3a). The reabsorbed vacuoles were related to the release of T_3_ and T_4_ [19]. This could be a negative feedback regulation caused by a decrease in the concentration of T_3_ and T_4_ in the lizard plasma under PCBs exposure. Those results of thyroid hormone concentrations and histopathological results are similar in the lizards treated intraperitoneally with a single dose of Aroclor 1242, 1254 and 1260, cytochrome P4501A and P4502B. All PCB congeners are metabolized into hydroxylated metabolites (OH-PCB) that are similar in thyroid hormones structure, resulting in interaction with TH receptors (TR) leading to disruption of the normal thyroid homeostasis [4,7].

In the PCBs exposure groups, a significant decrease in plasma concentrations of T_3_ and T_4_ led to a down-regulation of ORD II expression in the liver. This result indicated that in the lizard liver, thyroid receptor expression is directly regulated by the concentration of thyroid hormones in the plasma [18].

In the lizard *Podarcis siculus* liver, the 5′ORD (type II) is an important regulatory enzyme that catalyzes the conversion of T_4_ to T_3_ [36]. Compared with the control group, the expression of 5′ORD (type II) in the liver was significantly increased in all PCBs exposure groups. This suggested that there is a feedback between the conversion of T_4_ to T_3_ and T_3_ metabolism in the liver. The increase in 5′ORD (type II) alleviated the decrease in the concentration of thyroid hormone T_3_ in plasma. This implied that there was negative feedback regulation between the expression of 5′ORD (type II) and thyroid hormone concentration in plasma. This phenomenon indicated that PCBs exposure affected the pre-synthesis of plasma thyroid hormone, but the conversion in the liver of T_4_ to T_3_ mitigated this effect.

In addition, with respect to the control group, the T_3_ to T_4_ ratio in the four exposed groups showed a significant ratio increase (Appendix A). These results showed that up-regulation of 5′ORD (type II) promotes the transformation of T_4_ into T_3_ in the liver. This suggested that there was a competitive negative feedback adjustment relationship between the conversion of T_4_ toT_3_ and the expression of 5′ORD (type II) that alleviated the decrease in the concentration of thyroid hormones in plasma.

Another possible mechanisms of action of PCBs’ thyroid disrupting effects is correlated HPT axis activity, so, the association between thyroid activity and PCBs contamination in lizards may be related to an excess of TRH input with a decrease in TSH concentration from the pituitary gland. In fact, in the present study, in this species, the relationships PCBs-TRH depend on the level of contamination, we can assume that at higher concentrations of PCBs can stimulate the secretion of TRH with effect on the levels of thyroid-stimulating hormone (TSH) and therefore on the thyroid gland.

In summary, PCBs mainly affect the concentration of T_4_ in plasma and the transformation of T_4_ into T_3_ in the liver by increasing 5′ORD (type II). In particular, exposure to PCBs could affect iodine intake and use in the thyroid gland, resulting in thyroid insufficiency, negative feedback leading to thyroid epithelial hyperplasia and enlargement of follicular volume.

## 4. Materials and Methods

### 4.1. Experimental Design

Adult male specimens of *Podarcis siculus*, weighing 13–15 g, were live-captured in the neighborhood of Naples in May when the thyroid gland was in full functional activity [37] and then housed in large soil-filled terraria and indoor exposed to natural photoperiod and temperature. Lizards were fed daily on live fly larvae (*Tenebrio molitor*) and water dishes were always available in the terraria. Before starting the treatments, an acclimatization period of 20 days was allowed to reverse capture-related stress [36]. The experiments were performed in accordance with the ethical provisions imposed by the European Union and permitted by the National Committee of the Italian Ministry of Health on in vivo experimentation. After the period of acclimation, 45 animals were randomly assigned to three experimental terraria for 120 days. The soil for three experimental terraria was taken from 3 areas with different concentrations of PCBs from the Bagnoli brownfield area situated in the western part of the city of Naples (Campania region), Southern Italy. The concentrations of ∑PCB in the soil (5 kg) were 2.50 mg/kg (low-dose: Group A), 4.50 mg/kg (medium-dose: Group B) and 7.50 mg/kg (high-dose: Group C). The single 31 congeners in three groups of soil were summarized in Appendix A.

Another group (Group D, n = 15) was treated intraperitoneally with a single dose of 0.5 mg per kilogram of body weight of Aroclor 1242, 1254 and 1260, cytochrome P4501A and P4502B (PCB mixture) dissolved in 50 μL of corn oil and sacrificed 24 **h** after the injection. All substances were obtained from Caledon Laboratories Ltd. (Georgetown, Halton Hills, ON, Canada).

Untreated control lizards (n = 15) were housed for 120 days in nonpolluted terraria (time zero controls); a further control group (group treated with oil n = 15) was intraperitoneally injected with 50 μL of corn oil in a single dose and sacrificed 24 h after the injection.

Each animal was weighed once a month and weight changes were calculated. After 120 days, blood samples were collected by intracardiac puncture and put into heparinized tubes. Plasma used for the determination of hormonal levels was obtained by centrifuging blood samples for 10 min at 3500 rpm at 4 °C. After the collection of blood samples, animals were immediately anaesthetized by hypothermia and decapitated. Thyroid gland was removed for histopathological analysis, and the liver, kidney, brain, and skin were removed for biochemical analysis.

### 4.2. Biochemical Analysis

#### 4.2.1. Determination of 31 Congeners in Plasma and Tissues (Liver, Kidney, Brain, and Skin)

The plasma sample (50–100 μL) and homogenized tissues by gentleMACS^®^ (0.05–1.5 g) were subjected to liquid-liquid extraction after the phase precipitation protein, concentrated and loaded on the SPE column (solid-phase extraction) which retain the analytes of interest; PCBs were selected eluted with n-hexane. Agilent Technologies 6890 N MSD GC Gas Chromatograph 5973 performed the determination with Agilent Technologies 5973 ground detector. The limit of quantification for each congener varied among PCBs but was generally <0.01 ng/mL.

#### 4.2.2. Plasma Thyroid Hormones and TSH and TRH Assays

TRH and TSH levels were determined by immunoradiometric assay (IRMA) as previously reported in Sciarrillo et al. [18]. T_3_ and T_4_ levels were determined using radioimmunoassay (RIA) [18].

#### 4.2.3. Hepatic Thyroid Hormones and 5′ORD (Type II) Monodeiodinase

Livers were removed and flushed in a buffer composed of MOPS and EDTA at pH 7.4. The contents of T_3_ and T_4_ in hepatic tissue were determined by radioimmunoassay RIA and were expressed as ng/mg of tissue (fresh weight) [18]. The activity of the enzyme is expressed as pM T_3_/g (of liver)/h [18].

### 4.3. Histological Analysis

Thyroid glands were removed and immediately fixed in Bouin’s fixative and processed for light microscopy (LM). Serially cut paraffin sections (7 µm) were stained with Galgano stain and observation was performed using a Zeiss Axioskop microscope. The height of the follicular cells was measured in 30 cells every 3 slides and always on the second section of both normal and treated samples using a digital system of image (KS 300).

### 4.4. Statistical Analysis

The obtained data have been averaged prior to calculating the experimental group mean and the standard error of the mean. As revealed by the *x*^2^ test, data were not different from the normal distribution. The control and experimental data of all the groups were tested together for significance using two-way ANOVA, followed by Bonferroni’s for multi-group comparison and Student’s *t*-test for between-group comparison using GraphPad Prism version 8.00 for Windows, GraphPad Software, La Jolla, CA, USA. Differences were considered significant at *p <* 0.001.

## 5. Conclusions

This study showed that lizards could take in PCBs through dermal penetration when living in soil contaminated with PCBs. In summary, PCBs can disrupt the lizard ’s thyroid system. In particular, exposure to PCBs mainly decreased T_3_ and T_4_ levels in the blood of lizards and inhibited the activity of the thyroid gland, which showed similar results with lizards treated intraperitoneally with a single dose of Aroclor 1242, 1254 and 1260, cytochrome P4501A and P4502B. The potential mechanisms and modes of action of PCBs’ effects on the thyroid gland of the *Podarcis siculus* lizard occur at different levels including the central control system, the hypothalamic-pituitary-thyroid (HPT) axis, as well as hormonal bioavailability and metabolism in peripheral organs (Figure 5).

## Figures and Tables

**Figure 1 ijms-23-04790-f001:**
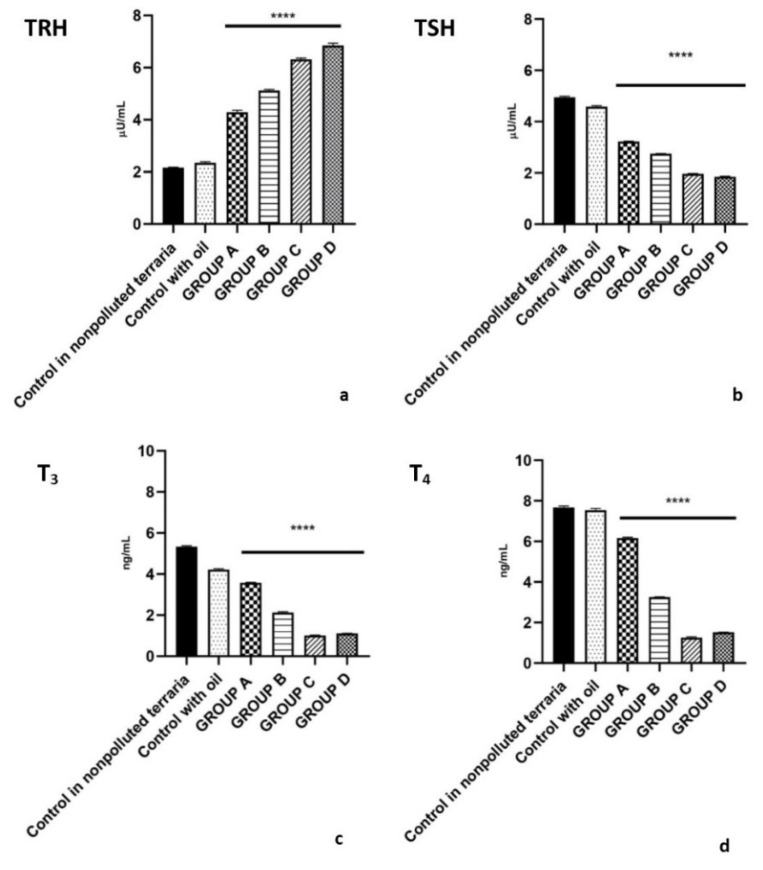
Plasma levels of TRH (**a**), TSH (**b**), T_3_ (**c**), T_4_ (**d**) after exposure to 2.50 (Group A), 4.50 (Group B),7.50 (Group C) mg/kg PCBs and after treatment with a single dose of Aroclor 1242, 1254 and 1260, cytochrome P4501A and P4502B (Group D) (**** *p* < 0.001, in the comparison with the different controls). A more detailed description is in the text.

**Figure 2 ijms-23-04790-f002:**
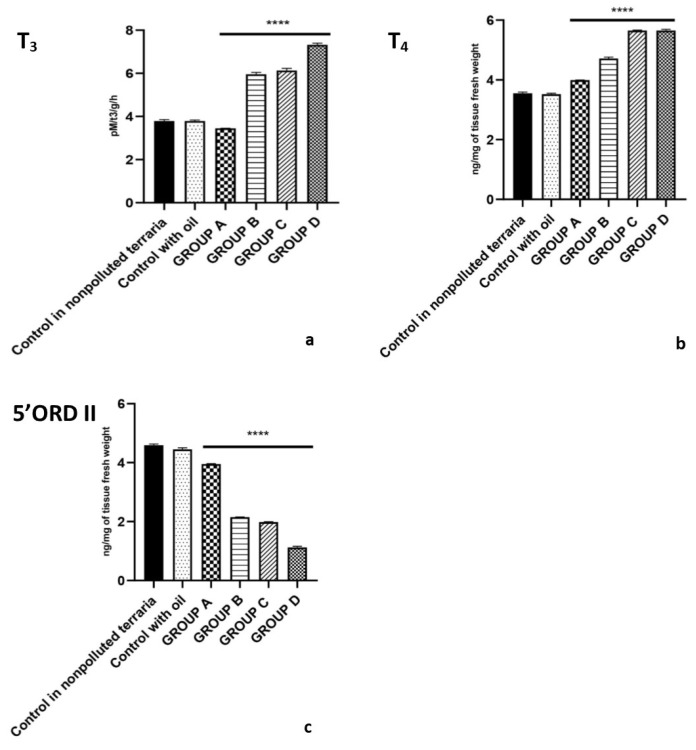
Hepatic content of T_3_ (**a**), T_4_ (**b**), 5′ORD II monodeiodinase activity (**c**) after exposure to 2.50 (Group A), 4.50 (Group B),7.50 (Group C) mg/kg PCBs and after treatment with a single dose of Aroclor 1242, 1254 and 1260, cytochrome P4501A and P4502B (Group D) (**** *p* < 0.001, in the comparison with the different controls). A more detailed description in the text.

**Figure 3 ijms-23-04790-f003:**
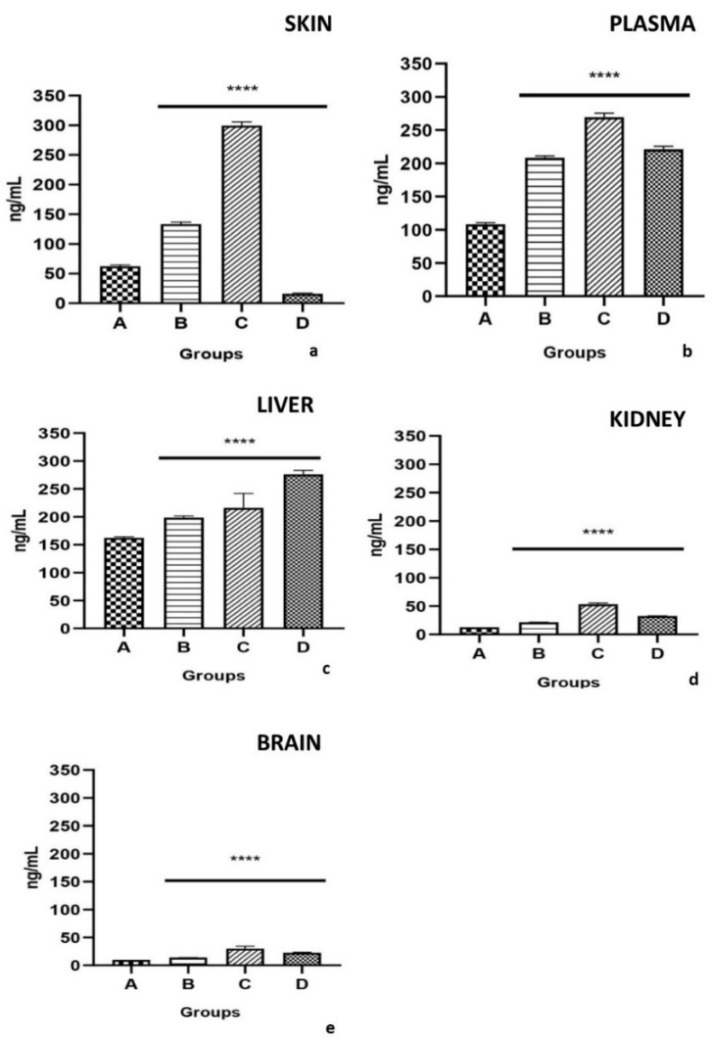
PCBs content in skin (**a**), plasma (**b**), liver (**c**), kidney (**d**) and brain (**e**) after exposure to 2.50 (Group A), 4.50 (Group B), 7.50 (Group C) mg/kg PCBs and after treatment with a single dose of Aroclor 1242, 1254 and 1260, cytochrome P4501A and P4502B (Group D) (**** *p* < 0.001, in comparison with the different controls). A more detailed description is in the text.

**Figure 4 ijms-23-04790-f004:**
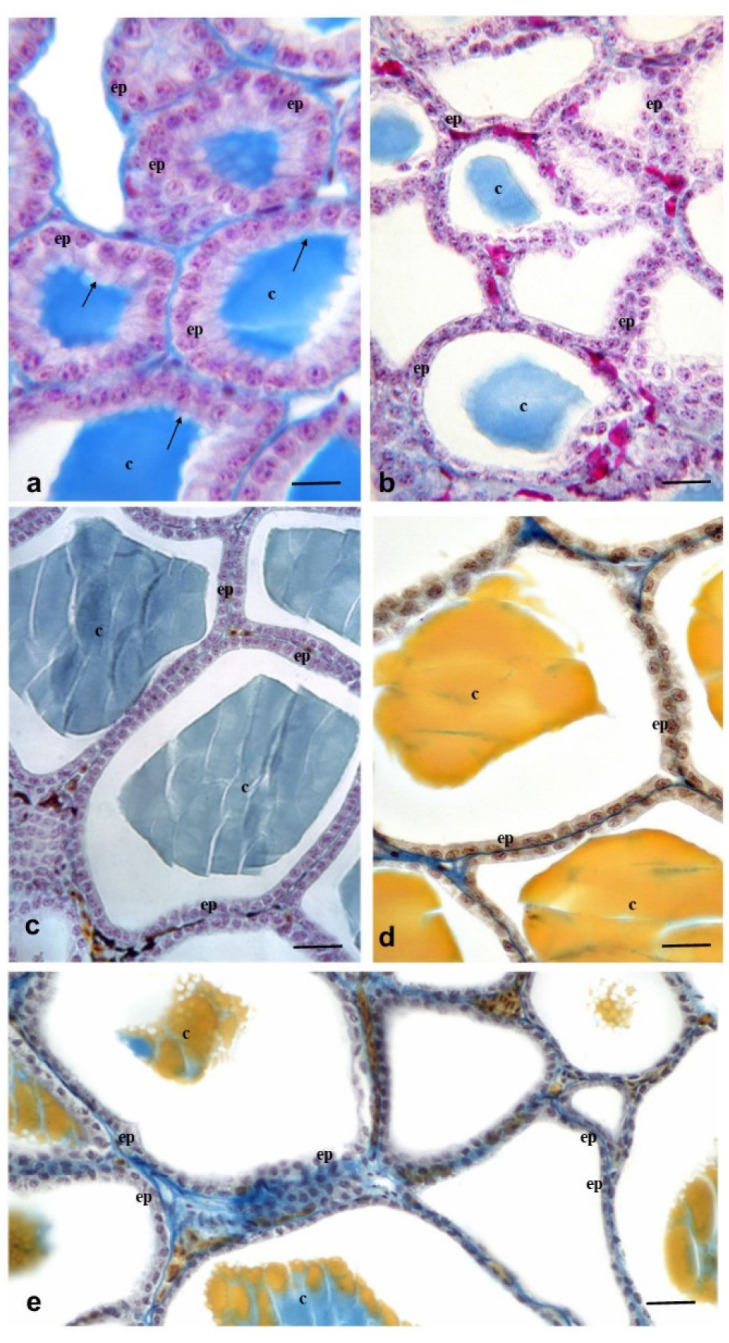
Thyroid gland of lizard *p. siculus* (stain Galgano I): scale bar: 20 μm. (**a**) Control lizards were housed for 120 days in nonpolluted terraria: the cuboidal follicular epithelial cells (ep), the colloid (c) and the reabsorption vacuoles (arrow) are shown; (**b**) lizards of the Group A: the follicular epithelium (ep) is lower than in control animals; (**c**) lizards of the Group B: the follicular epithelium (ep) is very low and the decrease in colloids (c) in follicles is very evident as compared to control animals; (**d**) lizards of the Group C: the follicular epithelium (ep) is lower than normal, but the colloid (c) is present in the follicles; (**e**) lizards of the Group D: the follicular epithelium (ep) is lower than normal and few reabsorption vacuoles are visible in the colloid (c).

**Figure 5 ijms-23-04790-f005:**
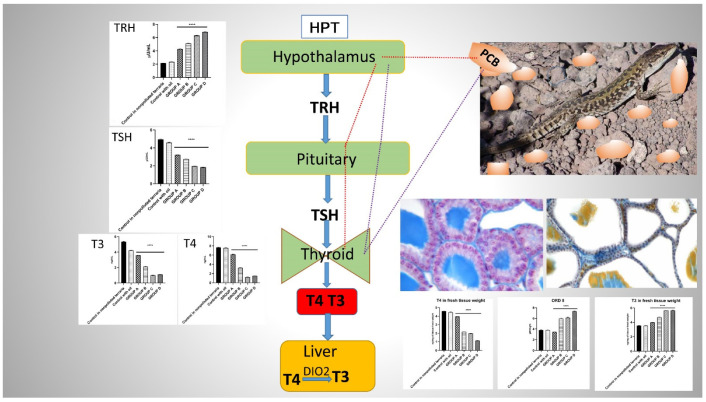
The potential mechanisms and modes of action of PCBs’ effects on the thyroid gland of the *Podarcis siculus* lizard.

**Table 1 ijms-23-04790-t001:** Mortality of specimens of *Podarcis siculus* after exposure to 2.50 (Group A), 4.50 (Group B), 7.50 (Group C) mg/kg (soil) PCBs and after treatment with a single dose of Aroclor 1242, 1254 and 1260, cytochrome P4501A and P4502B (Group D) in comparison with the different controls. A more detailed description in the text.

Treatments	Dead Animals (% of Animal Mortality)
Control in nonpolluted terraria	0
Control with oil	0
GROUP A	1 (6%)
GROUP B	2 (12%)
GROUP C	5 (33%)
GROUP D	5 (33%)

**Table 2 ijms-23-04790-t002:** Whole lizard weight after exposure to 2.50 (Group A), 4.50 (Group B), 7.50 (Group C) mg/kg PCBs and after treatment with a single dose of Aroclor 1242, 1254 and 1260, cytochrome P4501A and P4502B (Group D) in comparison with the different controls. Results are presented as mean ± standard error and the statistical analysis was performed separately for the five different periods (0, 30, 60, 90 and 120 days) (**** *p* < 0.001, in comparison with the different controls). A more detailed description is in the text.

Treatments	Whole Lizard Weight (g)				
	Day 0	Days 30	Days 60	Days 90	Days 120
Control in nonpolluted terraria	12 ± 0.01	13 ± 0.05	15 ± 0.05	17 ± 0.05	20 ± 0.05
Control with oil	12 ± 0.01	13 ± 0.04	15 ± 0.04	17 ± 0.05	20 ± 0.05
GROUP A	12 ± 0.02	12 ± 0.02	12 ± 0.04	11 ± 0.01 ****	10 ± 0.05 ****
GROUP B	12 ± 0.02	12 ± 0.05	11 ± 0.05	10 ± 0.05 ****	9 ± 0.05 ****
GROUP C	12 ± 0.02	11 ± 0.02	10 ± 0.05	9 ± 0.05 ****	8 ± 0.05 ****
GROUP D	12 ± 0.05	12 ± 0.01	10 ± 0.04	9 ± 0.05 ****	8 ± 0.01 ****

**Table 3 ijms-23-04790-t003:** Height of follicular epithelial cells after exposure to 2.50 (Group A), 4.50 (Group B), 7.50 (Group C) mg/kg PCBs and after treatment with a single dose of Aroclor 1242, 1254 and 1260, cytochrome P4501A and P4502B (Group D) in comparison with the different controls. Results are presented as mean ± standard error, and the statistical analysis was performed separately for the five different periods (0, 30, 60, 90 and 120 days). (**** *p* < 0.001, in the comparison with the different controls). A more detailed description is in the text.

Treatments	Height of Follicular Epithelium (µm)
Control in nonpolluted terraria	22.1 ± 0.02
Control with oil	21.3 ± 0.05
Group A	10.6 ± 0.03 ****
Group B	6.61 ± 0.01 ****
Group C	3.32 ± 0.05 ****
Group D	3.15 ± 0.04 ****

## Data Availability

Not applicable here.

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
