# Peer review of "Toxic Effects on Thyroid Gland of Male Adult Lizards (*Podarcis Siculus*) in Contact with PolyChlorinated Biphenyls (PCBs)-Contaminated Soil"

_ijms, 2022, doi:10.3390/ijms23094790_

Round 1

Reviewer 1 Report

Dear authors,

the manuscript "Toxic Effects on Thyroid Gland of Male Adult Lizards (Podarcis siculus) in contact To PolyChlorinated Biphenyls (PCBs)-Contaminated Soil" examines very important question. However, substantial improvements are required in order for this manuscript to be accepted for publication.

First of all, the language needs significant corrections, and I marked in yellow some of the phrases. Furthermore, this research is not put in the context of currently know data on skin absorption of toxins in lizards, and only a very few data on the toxicity of PBCs is mentioned in the introduction.

In the introduction, hypothesis was not clearly defined. In the results, no statistical data was provided (apart from several p values). Table 3 is mentioned in the text, but is not provided.

There are more things that need to be corrected, and I wrote it in the pdf.

Once all these data are provided, and manuscript corrected, it will be a valuable contribution to our knowledge on lizards' ecotoxicology.

Author Response

1. First of all, the language needs significant corrections, and I marked in yellow some of thephrases.

English has been revised throughout the text and linguistic corrections have been made as suggested

2.  Furthermore, this research is not put in the context of currently know data on skin absorption of toxins in lizards, and only a very few data on the toxicity of PBCs is mentioned inthe introduction.

The data currently known on the cutaneous absorption of toxins in lizards and in particular those concerning the toxicity of PBCs are very few this is the reason why the bibbliographic data mentioned in the introduction are very few.

3. In the introduction, hypothesis was not clearly defined.

The aims of the manuscript are been rewrited.

4. In the results, no statistical data was provided (apart from several p values).

5. Table 3 is mentioned in the text, but is not provided.

The table 3 has been inserted in the text.

6. There are more things that need to be corrected, and I wrote it in the pdf.

All linguistic correction have been made as suggested in the pdf.

Reviewer 2 Report

The work deals with a topic, contamination by polychlorinated biphenyls, widely dealt with in the literature. In mammals but also in low vertebrates. It, therefore, does not provide any new relevant information.   Furthermore, the tested concentrations are very high: present in a particularly polluted site, Bagnoli, they are unrealistic from an environmental point of view.   Last but not least, it is clear from the text that the animals were exposed to very high stress, as suggested by the 33% mortality. The work therefore also presents some weaknesses from an ethical point of view.    

Author Response

1. Furthermore, the tested concentrations are very high: present in aparticularly polluted site, Bagnoli, they are unrealistic from an environmental point of view.

Bagnoli brownfield site is characterized by soil multielement contamination. It is the second largest integrated

steelworks in Italy under remediation by a Government project. The industries, originally sited in the area and now closed, included a smelting plant, a very large integrated steelwork (with blast furnaces, coke ovens, sinter plants, electric arc steel furnace). 

2. Last but not least, it is clear from the text that the animals were exposed to very high stress, assuggested by the 33% mortality. The work therefore also presents some weaknesses from anethical point of view

The experiments were performed in accordance with the ethical provisions imposed by the European Union and permitted by the National Committee of the Italian Ministry of Health on in vivo experimentation.

Furthermore, the lizards at the Bagnoli site live naturally.

Round 2

Reviewer 1 Report

Dear authors,

many of the comments that I provided in the first round were left unanswered, e.g. whether you tested your data for normality; you did not provide statistical results (p value is not sufficient); there are few papers on  dermal exposure of lizards to different contaminants that are not mentioned and are important as they put your research into wider context...

Please, check once again my original review and answer all the points raised there.

English was also only partially improved, and it is still in some instances , quite bad (the meaning of some phrases is non-sense and thy are literally translated from Italian to English; e.g. Levels of hormones plasma --> hormones do not have plasma; it should be written plasma hormones- the phrase you use in the M&M but not in the results). The language of the manuscript needs to be corrected by a native (or proficient) speaker to avoid such phrases.

Author Response

Reviewer 1

  1. First of all, the language needs significant corrections, and I marked in yellow some of thephrases.

English has been revised throughout the text and linguistic corrections have been made as suggested.

2  Furthermore, this research is not put in the context of currently know data on skin absorption of toxins in lizards, and only a very few data on the toxicity of PBCs is mentioned inthe introduction.

A rich bibliography concerning skin absorption in reptiles of various contaminants has been included though

the data currently known on the cutaneous absorption of PBCs are very few this is the reason why the bibbliographic data mentioned in the introduction are very few.

3 In the introduction, hypothesis was not clearly defined.

The aims of the manuscript are been rewrited.

4 In the results, no statistical data was provided (apart from several p values).

The obtained data have been averaged prior to calculating experimental group mean and the standard error of the mean. As revealed by the x2 test, data was not different from normal distribution. The control and experimental data of all the groups were tested together for significance using two-way ANOVA, followed by Bonferroni’s for multi-group comparison and Student’s t test for between group comparison using GraphPad Prism version 8.00 for Windows, GraphPad Software, La Jolla California USA. Differences were considered significant at p < 0.001. Therefore, in the results have been inserted the statistical data.

  1. Table 3 is mentioned in the text, but is not provided.

The table 3 has been inserted in the text.

  1. There are more things that need to be corrected, and I wrote it in the pdf.

All linguistic correction have been made as suggested in the pdf.

  1. whether you tested your data for normality; you did not provide statistical results (p value is not sufficient);

The obtained data have been averaged prior to calculating experimental group mean and the standard error of the mean. As revealed by the x2 test, data was not different from normal distribution. The control and experimental data of all the groups were tested together for significance using two-way ANOVA, followed by Bonferroni’s for multi-group comparison and Student’s t test for between group comparison using GraphPad Prism version 8.00 for Windows, GraphPad Software, La Jolla California USA. Differences were considered significant at p < 0.001.

Therefore, in the results have been inserted the statistical data.

  1. there are few papers on dermal exposure of lizards to different contaminants that are not mentioned and are important as they put your research into wider context.

A rich bibliography concerning skin absorption in reptiles of various contaminants has been included though

the data currently known on the cutaneous absorption of PBCs are very few this is the reason why the bibbliographic data mentioned in the introduction are very few.

English was also only partially improved, and it is still in some instance , quite bad (the meaning of some phrases is non-sense and thy are literally translated from Italian to English; e.g. Levels of hormones plasma --> hormones do not have plasma; it should be written plasma hormones- the phrase you use in the M&M but not in the results). The language of the manuscript needs to be corrected by a native (or proficient) speaker to avoid such phrases.

The manuscript has been completely linguistically revised by a native speaker.

Reviewer 2 Report

Manuscript has been improved in clarity and English format but the critical issues presented by me have not been addressed. 

Author Response

Reviewer 2

  1. Furthermore, the tested concentrations are very high: present in aparticularly polluted site, Bagnoli, they are unrealistic from an environmental point of view.

Bagnoli brownfield site is characterized by soil multielement contamination. It is the second largest integrated steelworks in Italy under remediation by a Government project. The industries, originally sited in the area and now closed, included a smelting plant, a very large integrated steelwork (with blast furnaces, coke ovens, sinter plants, electric arc steel furnace). 

  1. Last but not least, it is clear from the text that the animals were exposed to very high stress, assuggested by the 33% mortality. The work therefore also presents some weaknesses from anethical point of view

The experiments were performed in accordance with the ethical provisions imposed by the European Union and permitted by the National Committee of the Italian Ministry of Health on in vivo experimentation. Furthermore, the lizards at the Bagnoli site live naturally.

Manuscript has been improved in clarity and English format but the critical issues presented by me have not been addressed.

The authors responded to both the first and second requests.
